# New Therapy Options for Neuroendocrine Carcinoma of the Pancreas—The Emergent Substance GP-2250 and Gemcitabine Prove to Be Highly Effective without the Development of Secondary Resistances In Vitro and In Vivo

**DOI:** 10.3390/cancers14112685

**Published:** 2022-05-29

**Authors:** Marie Buchholz, Johanna Strotmann, Britta Majchrzak-Stiller, Stephan Hahn, Ilka Peters, Julian Horn, Thomas Müller, Philipp Höhn, Waldemar Uhl, Chris Braumann

**Affiliations:** 1Department of General and Visceral Surgery, St. Josef-Hospital, Ruhr-University Bochum, 44791 Bochum, Germany; johanna.strotmann@rub.de (J.S.); britta.majchrzak@rub.de (B.M.-S.); ilka.peters@rub.de (I.P.); julian.horn@klinikum-bochum.de (J.H.); philipp.hoehn@rub.de (P.H.); waldemar.uhl@klinikum-bochum.de (W.U.); chris.braumann@kklbo.de (C.B.); 2Department of Molecular Gastrointestinal Oncology, Ruhr-University Bochum, 44780 Bochum, Germany; stephan.hahn@rub.de; 3Geistlich Pharma AG, 6110 Wolhusen, Switzerland; thomas.mueller@geistlich.com; 4Department of General, Visceral and Vascular Surgery, Evangelische Kliniken Gelsenkirchen, Akademisches Lehrkrankenhaus der Universität Duisburg-Essen, 45879 Gelsenkirchen, Germany

**Keywords:** neuroendocrine carcinoma, pancreas, chemotherapy, xenograft, secondary resistance

## Abstract

**Simple Summary:**

Neuroendocrine carcinoma of the pancreas is a highly aggressive form of neuroendocrine tumor associated with poor survival and increasing occurrence. GP-2250 is an emergent substance showing antineoplastic properties, especially in combination with Gemcitabine. This study was the first to evaluate the antineoplastic effects of GP-2250 on pancreatic neuroendocrine carcinoma. The combination of GP-2250 and Gemcitabine showed highly synergistic effects in a cell culture model, as well as in mice, without the development of secondary resistances. These findings form the basis for further clinical evaluation of a highly promising combination therapy.

**Abstract:**

Neuroendocrine carcinoma of the pancreas (pNEC) is an aggressive form of neuroendocrine tumor characterized by a rising incidence without an increase in survival rates. GP-2250 is an oxathiazinane derivate possessing antineoplastic effects, especially in combination with Gemcitabine on the pancreatic adenocarcinoma. The cytotoxic effects of the monotherapy of GP-2250 (GP-2250_mono_) and Gemcitabine (Gem_mono_), as well as the combination therapy of both, were studied in vitro using an MTT-assay on the QGP-1 and BON-1 cell lines, along with in vivo studies on a murine xenograft model of QGP-1 and a patient-derived xenograft model (PDX) of Bo99. In vitro, Gem_mono_ and GP-2250_mono_ showed a dose-dependent cytotoxicity. The combination of GP-2250 and Gemcitabine exhibited highly synergistic effects. In vivo, the combination therapy obtained a partial response in QGP-1, while GP-2250_mono_ and Gem_mono_ showed progressive disease or stable disease, respectively. In Bo99 PDX, the combination therapy led to a partial response, while the monotherapy resulted in progressive disease. No development of secondary resistances was observed, as opposed to monotherapy. This study was the first to evaluate the effects of the emerging substance GP-2250 on pNEC. The substance showed synergism in combination with Gemcitabine. The combination therapy proved to be effective in vitro and in vivo, without the development of secondary resistances.

## 1. Introduction

Neuroendocrine tumors (NET) are a highly heterogenic group of neoplasms originating from the neuroendocrine system [1] and can affect various organ systems. The most common sites of emergence are the lungs and the gastroenteropancreatic system [2,3,4]. Pancreatic neuroendocrine tumors (pNET), in general, are rather unusual entities among pancreatic neoplasms [2,4,5]; however, there has been a marked rise in their incidence over the past decade [2,4,5,6], while the overall survival rates have not increased [6,7].

According to the fourth edition of the World Health Organization’s classification of endocrine tumors, NETs are graded (Grade 1–3) depending on the mitotic count and Ki-67 labelling index [8]. While Grade 1 and 2 are characterized by overall favorable survival rates [2,6,7,9,10], especially compared to other pancreatic neoplasms [11], Grade 3, the so-called neuroendocrine carcinoma (NEC) [8], exhibits highly malignant traits and a significantly poorer prognosis [9,10].

The gold standard of NEC treatment is oncologic resection. Concerning pancreatic NEC (pNEC), this involves, depending on the localization, pancreatic head resection, Whipple’s procedure, distal pancreatic resection or even total pancreatectomy, ranking among the most complex procedures in abdominal surgery [12]. However, the majority of patients present metastasis at the time of diagnosis [10,13] and therefore cannot be subjected to surgery [14,15]. In these cases, the first line therapy consists of Carboplatin or Cisplatin in combination with Etoposide [15]. Side effects are common and severe [16]. Concerning the platinum analogues, Carbo- or Cisplatin adverse effects include nephro-, hepato- and ototoxicity, with neurotoxicity constituting the most important dose-limiting problem. This encompasses the persisting to permanent loss of taste and position and vibration sense, weakness, tremor or even leukoencephalopathy and seizures, massively infringing the patient’s quality of life [17,18]. As for Etoposide, a semisynthetic epipodophyllotoxin, myelosuppresion is the dose-limiting toxicity resulting in pancytopenia. Further side effects include gastrointestinal toxicity, asthenia, alopecia, fever and chills [19]. An adequate performance status is required prior to application [16]. Hence, the development and research of new therapy regimen, especially regarding emerging substances, is indispensable to improve the survival and quality of life in these highly vulnerable patients.

The oxathiazinane derivate GP-2250 (1,4,5-oxathiazan-dioxide-4,4) presents a recent development and has been demonstrated to possess antiproliferative, antineoplastic and migration inhibiting effects on pancreatic tumor cells in vitro [20]. It has a six-ring structure with oxygen, sulfurdioxide and nitrogen atoms on the positions 1, 4 and 5, respectively. The exact mechanism of action is yet to be fully explored; the findings of our research group suggest the induction of cell death via the increased release of reactive oxygen species as well as mitochondrial dysfunction [20,21,22]. In vivo, GP-2250 shows the reduction of tumor growth in the xenografts of established pancreatic cancer cell lines, as well as in patient-derived xenograft (PDX) models of pancreatic adenocarcinoma, accompanied by rarely occurring side effects [20]. The evaluation of the maximal tolerable dose revealed acute toxicity at 2000 mg/kg*BW and chronic toxicity at concentrations higher than 1000 mg/kg*BW in nude mice. No changes in body weight or vital functions were observed at lower concentrations [20,22].

So far, no representative studies have been published analyzing the anti-neoplastic effect of GP-2250 in pNEC. Our study was designed to gain more information about the novel therapeutic regime in this rare tumor entity, especially as further therapeutic options are urgently needed. The work is focused on the anti-neoplastic effect of GP-2250 in combination with Gemcitabine in vitro and in vivo.

## 2. Materials and Methods

### 2.1. Cells and Tissue

In this study, the established human pancreatic endocrine cell lines QGP-1 (AcceGen Biotech Cat.# ABC-TC0918) and BON-1 (kindly provided by PD Dr. med. Jörg Schrader, University Medical Center Hamburg-Eppendorf, Hamburg, Germany) were used. After reception from the cell bank, the cells were passaged for less than 6 months, and authentication was performed via short tandem repeat analysis. The QGP-1 cells were cultured in Dulbecco’s Modified Eagle Medium (DMEM) supplemented with 10% fetal calf serum (FCS) premium, penicillin (100 U/mL), streptomycin (100 U/mL) and 2 mM L-Glutamine (each PAN-Biotech, Aidenbach, Germany). The BON-1 cells were cultured in DMEM/F-12 supplemented with 10% FCS premium, penicillin (100 U/mL), streptomycin (100 U/mL) and 2 mM L-Glutamine (each PAN-Biotech, Aidenbach, Germany).

The cells were grown as a monolayer and cultured in 10 cm dishes at 37 °C and 5% CO_2_ in a humidified atmosphere.

For the in vivo patient-derived xenograft (PDX) experiments, human pancreatic tissue Bo99 was used. This tissue was derived from a neuroendocrine carcinoma of the pancreas head, pT3 pN1 (4/15) M0 L1 V1 Pn1, UICC Stadium IIb, G3. This patient postoperatively received six cycles of Cisplatin and Etoposide and developed tumor relapse 6 months afterwards. The treatment with Carboplatin and Etoposide was initiated and modified. Eventually, chemotherapy was terminated due to the deterioration of the patient’s general condition followed by exitus letalis 23 months after the initial diagnosis.

### 2.2. Reagents

GP-2250 ultrapure powder (Geistlich Pharma AG, Wolhusen, Switzerland) was set to physiological pH after being dissolved in double distilled water (ddH_2_O) and subsequently sterile filtered. The preparation was freshly performed once weekly.

### 2.3. MTT

In total, 1.5 × 10^4^ QGP-1 or BON-1 cells per well were seeded in 96-well plates and incubated for 24 h for the acquisition of a sub-confluent monolayer. Subsequently, the QGP-1 cells were treated with different concentrations of GP-2250 (100 µM, 150 µM, 175 µM, 200 µM and 300 µM) and Gemcitabine (0.001 µM, 0.01 µM, 0.1 µM, 1 µM and 10 µM) and incubated for 48 h to determine dose-response. Additionally, combinations (100 µM GP-2250 + 0.001 µM Gemcitabine, 100 µM GP-2250 + 0.01 µM Gemcitabine, 175 µM GP-2250 + 0.001 µM Gemcitabine, 175 µM GP-2250 + 0.01 µM Gemcitabine, 200 µM GP-2250 + 0.001 µM Gemcitabine, 200 µM 2250 + 0.01 µM Gemcitabine) were analyzed.

In the BON-1 cells, the dose response was determined similarly to the QGP-1 cells, and combinations of 25 µM, 50 µM and 100 µM GP-2250 with 100 µM and 1000 µM Gemcitabine were analyzed.

A total of 2 h prior to measurement, 10 µL yellow MTT (3-(4,5-Dimethylthiazol-2-yl)-2,5-diphenyltetrazoliumbromid) reagent (5 mg/mL) was applied into each well. Viable cells metabolize yellow MTT into violet Formazan crystals; thus, the number of viable cells is directly proportional the amount of violet Formazan. The test media was removed, and 100 µL DMSO (Dimethylsulfoxide) was added. After the complete solution of the crystals, the viability was analyzed via a microplate absorbance reader measuring the optical density at a wavelength of 550 nm and at 720 nm as reference (UVM 340, Biochrom, Cambridge, UK). The assay was repeated in four to six independent experiments in consecutive passages. The MTT assay was performed as previously described by van Meerloo et al. [23].

### 2.4. Xenograft QGP-1 and Patient Derived Xenograft Bo99

For this study, 6-week old female NMRI Foxn1nu/Foxn1nu mice (Janvier, Saint-Berthevin, France) were used. One week prior to initiation, they were acclimated into a 12 h light cycle-controlled environment. Standard laboratory food and water were allowed ad libitum. Anaesthesia was performed by Isofluran per inhalationem. As described previously, either 5 × 10^6^ QGP-1 [20,24] cells, or tumor tissue fragments in the case of Bo99 [25], were implanted subcutaneously into each flank region. Afterwards, the mice were monitored for general health status and the development of subcutaneous tumors. The tumor volume was calculated as V = ½(ab)^2^ (a = larger axis, b = smaller axis and a⊥b) via the measurement of the tumor diameter using a caliper. After reaching a tumor volume of 200 mm^3^, the mice were randomized into several groups: Group 1, controls (*n* = 6); group 2, GP-2250 monotherapy (*n* = 6); group 3, Gemcitabine monotherapy (*n* = 6); group 4, combination therapy GP-2250 and Gemcitabine (*n* = 6). In Bo99, GP-2250 monotherapy was not performed. The controls received a physiological saline solution, group 2 was intraperitoneally treated with 500 mg/kg*BW GP-2250 three times per week, group 3 was intraperitoneally treated with 50 mg/kg*BW Gemcitabine twice weekly and group 4 intraperitoneally received 500 mg/kg*BW 2250 and 50 mg/kg*BW Gemcitabine thrice and twice weekly, respectively, on alternating days. The tumor volume was measured twice a week. The experiment was terminated when the tumor volume reached 1000 mm^3^. The tumors were evaluated after 4 and 6 weeks according to the Response Evaluation Criteria in Solid Tumors (RECIST) [3].

To analyze the development of a secondary resistance against the combination of GP-2250 and Gemcitabine a therapeutical break was performed after 60 days of treatment. The combination therapy was started once again after the initial volume was reached. The above-mentioned scheme was used again. Due to a primary resistance, the Gemcitabine monotherapy was not interrupted.

### 2.5. Ethics Approval and Consent to Participate

The local ethical committee approved the collection of sample tissue from patients with pancreatic cancer as well as the implantation and expansion of cancer tissue in xenograft mouse models. The written, informed consent of all patients was documented according to the local ethics guidelines. The study was conducted according to the Declaration of Helsinki. All procedures were performed according to a protocol approved by the Ethics Committee of the Ruhr-University Bochum, Germany (permission no 2392, 10th amendment). All of the animal experiments were performed according to the guidelines of the local Animal Use and Care Committees (permission no. 81-02.04.2018.A169).

### 2.6. Statistical Analysis

The results of the MTT assay (percentage of viable cells), as well as the characteristics of mice (body weight, tumor volume), are presented as mean ± standard derivation. The hypotheses of no difference between the four groups for the continuous variables were tested using ANOVA, and pairwise tests were performed using *t* tests. *p* values ≤ 0.05 were considered to be statistically significant and were indicated in the figures as follows: *** *p* ≤ 0.001, ** *p* ≤ 0.01, * *p* ≤ 0.05. The testing was performed using Graph Pad Prism 9.1.0 by Graph Pad Software, San Diego, CA, USA.

## 3. Results

### 3.1. MTT

As previously described, GP-2250 has a dose-dependent effect on the cell viability of pancreatic adenocarcinoma [20]. In order to determine the effects and potential synergism of GP-2250 in combination with Gemcitabine on pancreatic neuroendocrine tumors, MTT assays were performed. In QGP-1, concerning GP-2250, a significant reduction in cell viability was achieved in comparison to the control, starting at concentrations of 100 µM, with an exemption of 150 µM. On average, incubation with 100 µM and 150 µM GP-2250 led to a decrease in cell viability of 7% ± 0.069 (*p* = 0.038) and 5% ± 0.113 (*p* = 0.147), and the application of 200 µM, 250 µM and 300 µM reduced cell viability to 35% ± 0.080, 50% ± 0.124 and 5% ± 0.033 (*p* < 0.001), respectively (Figure 1).

In Gemcitabine, a significant reduction in the cell viability in the QGP-1 cell line was observed at concentrations of 0.1 µM and higher. The application of 0.001 µM and 0.01 µM did not significantly impact cell viability, whereas at concentrations of 0.1 µM, 1 µM and 10 µM, 46% ± 0.054, 28% ± 0.071 and 13% ± 0.047 (*p* < 0.001) of cells were viable (Figure 1).

Similarly, the dose response to Gemcitabine and GP-2250 was determined in BON-1 cells (Figure 2).

Combining GP-2250 and Gemcitabine led to a highly significant synergism in the reduction in cell viability in both the QGP-1 and BON-1 cells.

Whereas in QGP-1, the concentration of 100 µM GP-2250 and 0.001 µM or 0.01 µM Gemcitabine monotherapy led to no or only a slight reduction in cell viability (*p* = 0.953, *p* = 0.014, *p* = 0.096), the combination therapy proved to be highly effective for the combinations of 100 µM GP-2250 and 0.01 µM Gemcitabine (reduction in cell viability by 27% ± 0.075), 175 µM GP-2250 and 0.001 µM Gemcitabine (cell viability reduction by 39% ± 0.080), 175 µM GP-2250 and 0.01 µM Gemcitabine (cell viability reduction by 56% ± 0.083), 200 µM GP-2250 and 0.001 µM Gemcitabine (cell viability reduction 86% ± 0.091) and 200 µM GP-2250 and 0.01 µM Gemcitabine (cell viability reduction by 87% ± 0.084), *p* < 0.0001, each. The combination of 100 µM GP-2250 with 0.01 µM Gemcitabine as well as the combination of 175 µM GP-2250 with 0.001 µM Gemcitabine or 0.01 µM Gemcitabine reduced cell viability extremely significantly when compared to 100 µM or 175 µM GP-2250_mono_, respectively. This was also observed when combining 200 µM GP-2250 and 0.001 µM or 0.01 µM Gemcitabine, respectively, which proved to decrease cell viability significantly in comparison to 200 µM GP-2250_mono_.

In the BON-1 cells, neither monotherapy with 25 µM, 50 µM or 100 µM GP-2250 nor 100 µM or 1000 µM Gemcitabine led to a significant reduction in cell viability compared to the untreated control. However, the combination therapy of 100 µM GP-2250 and 100 µM Gemcitabine or 1000 µM Gemcitabine and 50 µM or 100 µM GP-2250, respectively, proved to be highly effective. Cell viability reductions of 35% ± 0,06; 46% ± 0,09 and 13% ± 0,03, respectively, were achieved (*p* < 0.0001 each) (Figure 3).

The cultivation of Bo99 was not successful in a monolayer cell culture.

### 3.2. PDX and Xenograft

In order to elucidate the therapeutic effects in vivo, the treatment was tested in the PDX model Bo99 and in xenografts derived from the established cancer cell line QGP-1.

In Bo99, the treatment with Gemcitabine and the combination of GP-2250 and Gemcitabine both led to a significant decrease in tumor growth volume in comparison to the controls (Figure 4). All of the controls showed progressive disease according to the RECIST (mean increase in tumor volume: 8.6 fold ± 4.094) and reached the abortion criteria within 6 weeks and therefore had to be terminated. The treatment with Gemcitabine mono reduced tumor growth in comparison to the control group; however, the tumor volume increased 1.77 fold ± 0.385 within 8 weeks and therefore showed progressive disease according to the RECIST. The combination therapy of GP-2250 and Gemcitabine, on the other hand, resulted in a partial response after 8 weeks, with a mean reduction to 50.5% ± 0.145 of the initial tumor volume. The combination therapy therefore proved to be highly effective in comparison to Gemcitabine mono (*p* < 0.001).

In order to further elucidate the development of secondary resistances against the therapy, a therapeutic break was introduced in the combination therapy cohort after 60 days of treatment but not in the Gemcitabine mono cohort, due to the development of primary resistance in the latter (Figure 5). The treatment of the combination cohort was resumed when the tumor reached the initial volume after two weeks. The renewed application of Gemcitabine and GP-2250 led to a decrease in tumor volume, resulting in a partial response again. The observed reduction under combination therapy was significant when compared to the tumor volume increase and therefore the progressive disease under Gemcitabine mono (*p* = 0.007).

Similarly, in QGP-1 (Figure 6), the controls exhibited a pronounced increase in tumor volume, which nearly doubled in size over the observation period (increase by 188% ± 0.413). The monotherapy with GP-2250 resulted in an average increase in tumor volume by 25.4% ± 0.334 in progressive disease as well. The treatment with Gemcitabine alone decreased tumor volume significantly compared to the untreated control (relative reduction in tumor volume: 22.4% ± 0.265, *p* = 0.007) and led to a stable disease according to the RECIST. However, only the combination of GP-2250 and Gemcitabine reduced the tumor volume to an extent that met the criteria for partial response according to the RECIST [3] (77.9% ± 0.091, *p* = 0.002, respectively). This presents a significant decrease in the tumor volume compared to Gemcitabine mono (*p* ≤ 0.001). A follow-up period with a therapeutical break was not feasible due to the tumor’s tendency to form necrosis and thus present abortion criteria.

## 4. Discussion

The data given are the first to evaluate the effects of the emerging substance GP-2250 on pNEC. The substance showed strong synergism in combination with Gemcitabine, and the combination therapy proved to be highly effective in vitro on two cell lines of pNEC and in vivo in a cell line model and a PDX model. The latter is an important model in translational medicine, bridging the gap between animal and human studies [26].

pNECs are a highly aggressive subgroup of neuroendocrine tumors tied to an extremely unfavorable prognosis [9,10]. The current first-line therapy, consisting of Cisplatin/Carboplatin in combination with Etoposide, is accompanied by poor tolerability and severe side effects [16] and requires a good performance status for application [15]. A second-line therapy has yet to be fully established. The German guidelines express an open preference for Capecitabine plus Temozolomid, FOLFOX (folinic acid, 5-Fluorouracil, Oxaliplatin) or FOLFIRI (folinic acid, 5-Fluorouracil, Irinotecan) [15], which, in turn, depends on the sufficient performance status of the patients as well [27]. The only targeted therapy that has been FDA approved in the last 30 years is the mTor inhibitor Everolimus, which has shown a median progression-free survival rate of 11 months in the Everolimus group compared to 4.6 months in the placebo group [28,29,30].

This underlines the crucial need to establish new therapy regimens and concepts with better tolerability, especially for frailer or more comorbid patients.

In the selection of the therapeutic agents for this study, we focused on agents applicable as a second-line therapy. The clinical course of our PDX model Bo99 indicates a necessary second-line therapy with moderate side effects. The first experiments in pancreatic adenocarcinoma suggest a synergistic effect of the novel agents GP-2250 and Gemcitabine [21].

Gemcitabine is a reputable chemotherapeutic agent that is approved for, among other things, locally advanced and/or metastasized ductal adenocarcinoma of the pancreas [31]. It is mainly applied in patients whose performance status does not allow for treatment with other agents [32]. Concerning neuroendocrine tumors, Gemcitabine has been systematically evaluated nearly exclusively in small collectives with small-cell lung cancer (SCLC). Cormier et al. observed a response rate of 29% in 29 patients with previously untreated SCLC [33], while van der Lee et al. found an overall response rate of 13% in a collective of 41 patients with pre-treated, limited- or extensive-stage SCLC [34]. In 2004, Kulke and colleagues examined a collective of 18 patients with metastatic neuroendocrine tumors, of which seven presented pancreatic primaries, and found disease stabilization in 65% of them following monotherapy with Gemcitabine, although no radiological or biochemical response was achieved [35]. All of the studies concurringly found Gemcitabine to be well tolerated, with a favorable side-effects profile [33,34,35]. Concerning pNEC in particular, data are currently lacking; however, several case-reports observed a response and satisfactory tolerability to treatment with Gemcitabine as a salvage therapy [36], third-line therapy [37] or in combination with S-1 [38,39].

The oxathiazinane derivate GP-2250, on the other hand, has been proven to possess antiproliferative and antineoplastic properties on pancreatic cancer in vitro and in vivo [20]. The tolerability of the combination therapy with Gemcitabine has been established as favorable; correspondingly, no test animal had to be terminated due to any adverse effects of the treatment [20]. Currently, the combination therapy is being tested in a Phase I/II trial in subjects with advanced pancreatic cancer [40].

This study in pNEC supports the major relevance of this combination. While the monotherapy with Gemcitabine achieved stable disease in a PDX model and progressive disease in QGP-1 cells, the combination therapy of GP-2250 and Gemcitabine even led to partial response according to the RECIST and thereby led to a significant decrease in tumor volume in comparison to the monotherapy. The number of PDX studies, especially for pNET, is quite small. Chetser et al. showed just a stable disease in a pNET PDX model using the FDA approved agent Everolimus and Sapanisertib, another mTor inhibitor [41]. In another study, the traditional Chinese drug baicalein induced apoptosis and protein changes in vitro and inhibited the migration and tumor growth of BON-1 in vivo [42].

Furthermore, after the therapeutic break, the treatment with the combination therapy resulted in a renewed response, while the monotherapy with Gemcitabine led to the development of resistances after 30 days of treatment. Chemoresistance to Gemcitabine is a well-described problem, developing within weeks after initiation [43]. The mechanisms promoting resistances against Gemcitabine are multifaceted; mediation via pathways including p53 [44], NF-κB, Akt/PI3K [45], reactive oxygen species [46], heat shock proteins [47] and micro RNAs [48] was identified. Overcoming this hurdle presents an enormous opportunity towards the improvement of both the survival and quality of life of this highly vulnerable collective of patients.

This study has several limitations. As in every PDX model, the direct extrapolation of the results to humans is not possible. Furthermore, the investigation is limited due to the absent control with the first-line therapy. In the clinical course, however, the patient showed recurrence and progress during this treatment, which ultimately had to be terminated due to poor tolerability. Moreover, based on the rare tumor entity, there is a limited number of cell lines accessible.

The combination of GP-2250 and Gemcitabine proved to be effective for the treatment of an established cell line and a PDX model of pNEC in vitro and in vivo, without the development of secondary resistances. Further studies are needed to comprehensively evaluate the effects of GP-2250 and Gemcitabine in neuroendocrine carcinoma, especially in auxiliary cell lines.

The present study forms the basis for future clinical trials of a promising combination therapy.

## 5. Conclusions

The present study was the first to evaluate the effects of the emerging substance GP-2250 on pNEC. The substance showed strong synergism in combination with Gemcitabine, and the combination therapy proved to be highly effective in vitro and in vivo on both an established cell line and a PDX model of pNEC. Additionally, no development of secondary resistances was observed for the combination treatment. These results are consistent with the findings of prior studies involving PDAC cells and concerning the combination therapy with Gemcitabine and GP-2250 [22]. The additional analysis of GP-2250 may be valuable not only in PDAC but also in pNEC. Although the transfer of animal study results to humans is limited, this study forms the basis for further clinical evaluation of a highly promising combination therapy.

## Figures and Tables

**Figure 1 cancers-14-02685-f001:**
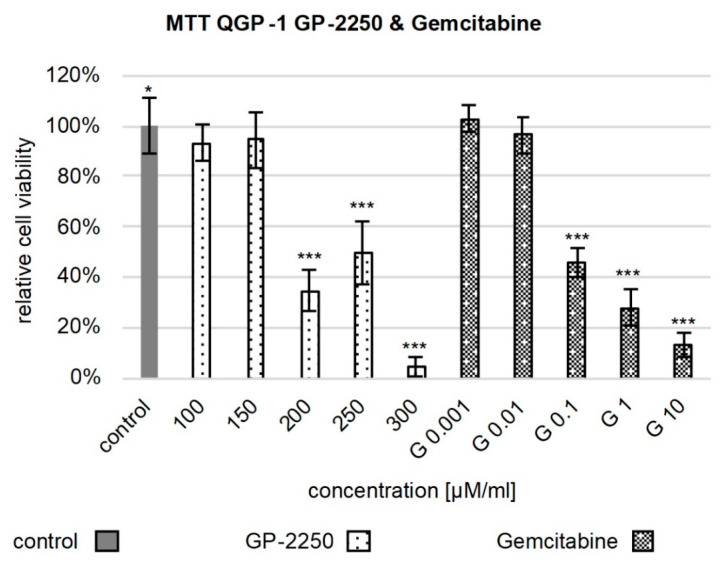
Cell viability assay of QGP-1 monotherapy. Figure 1 shows the reduction in cell viability observed in the MTT assay after treatment with 100 µM GP-2250, 150 µM GP-2250, 200 µM GP-2250, 250 µM GP-2250, 300 µM GP-2250, 0.001 µM Gemcitabine, 0.01 µM Gemcitabine, 0.1 µM Gemcitabine, 1 µM Gemcitabine and 10 µM Gemcitabine compared to the untreated control. Error bars show the standard deviation. *p*-values are indicated as follows: *** *p* ≤ 0.001, * *p* ≤ 0.05.

**Figure 2 cancers-14-02685-f002:**
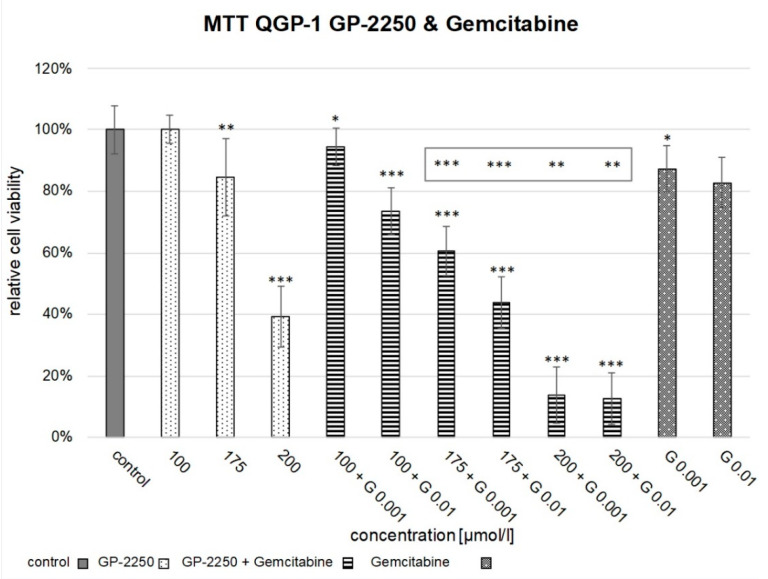
Cell viability assay of combination therapy in QGP-1. Figure 2 shows the reduction in cell viability observed in the MTT assay of the combination therapy of 100 µM GP-2250 + 0.001 µM Gemcitabine, 100 µM GP-2250 + 0.01 µM Gemcitabine, 175 µM + 0.001 µM Gemcitabine, 175 µM GP-2250 + 0.01 µM Gemcitabine, 200 µM GP-2250 + 0.001 µM and 200 µM GP-2250 + 0.01 µM Gemcitabine in comparison to the untreated controls and the monotherapy with 100 µM GP-2250, 175 µM GP-2250, 200 µM GP-2250, 0.001 µM Gemcitabine and 0.01 µM Gemcitabine. Error bars show the standard deviation. *p*-values are indicated as follows: *** *p* ≤ 0.001, ** *p* ≤ 0.01, * *p* ≤ 0.05. *p*-values above each column indicate the levels of significance in comparison to the untreated controls. *p*-values in the grey frame indicate the levels of significance of the combination compared to each single concentration in the monotherapy.

**Figure 3 cancers-14-02685-f003:**
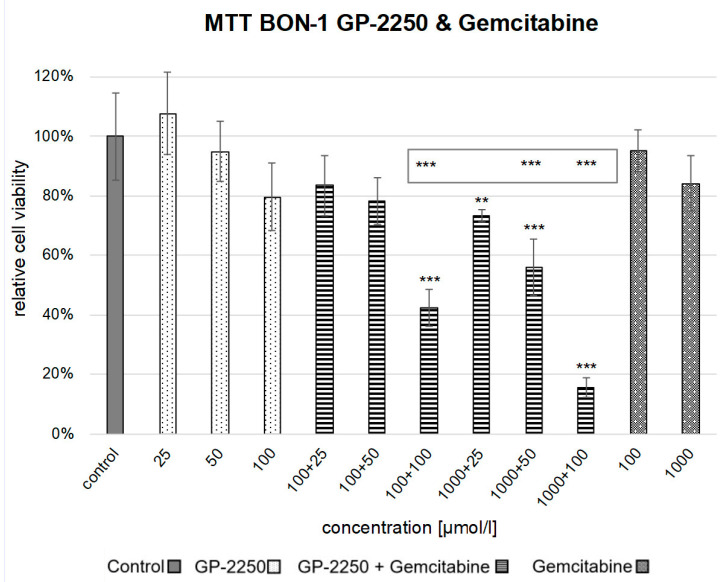
Cell viability assay of the combination therapy in BON-1. Figure 3 shows the reduction in cell viability observed in the MTT assay of the combination therapy with 25 µM GP-2250 + 100 µM Gemcitabine, 50 µM GP-2250 + 100 µM Gemcitabine, 100 µM GP-2250 + 100 µM Gemcitabine and 25 µM GP-2250 + 1000 µM Gemcitabine, 50 µM GP-2250 + 1000 µM Gemcitabine and 100 µM GP-2250 + 1000 µM Gemcitabine in comparison to the untreated controls and the monotherapy with 25 µM GP-2250, 50 µM GP-2250, 100 µM GP-2250, 100 µM Gemcitabine and 1000 µM Gemcitabine. Error bars show the standard deviation. *p*-values are indicated as follows: *** *p* ≤ 0.001, ** *p* ≤ 0.01. *p*-values above each column indicate the levels of significance in comparison to the untreated controls. *p*-values in the grey frame indicate the levels of significance of the combination compared to each single concentration in the monotherapy.

**Figure 4 cancers-14-02685-f004:**
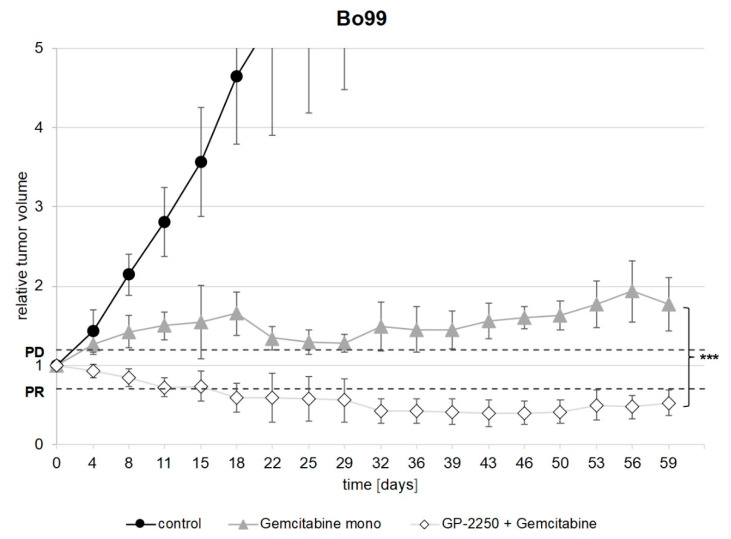
Development of the relative tumor volume of Bo99 during the observation period. Figure 4 shows the development of the relative tumor volume during the observation period of Bo99 tumors treated with Gemcitabine mono and a combination of Gemcitabine and GP-2250 in comparison to the untreated controls. PD, progressive disease; PR, partial response. Error bars show the standard deviation. *p*-values are indicated as follows: *** *p* ≤ 0.001.

**Figure 5 cancers-14-02685-f005:**
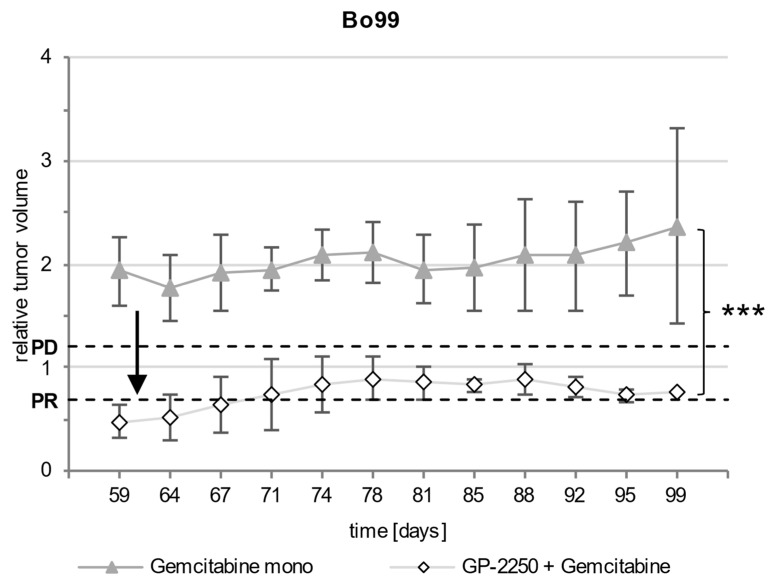
Development of the relative tumor volume during the follow-up period. Figure 5 shows the development of the relative tumor volume during the follow-up period of Bo99 tumors treated with Gemcitabine mono and a combination of Gemcitabine and GP-2250. The arrow indicates the start of the therapeutic break of the combination cohort. PD, progressive disease; PR, partial response. Error bars show the standard deviation. *p*-values are indicated as follows: *** *p* ≤ 0.001.

**Figure 6 cancers-14-02685-f006:**
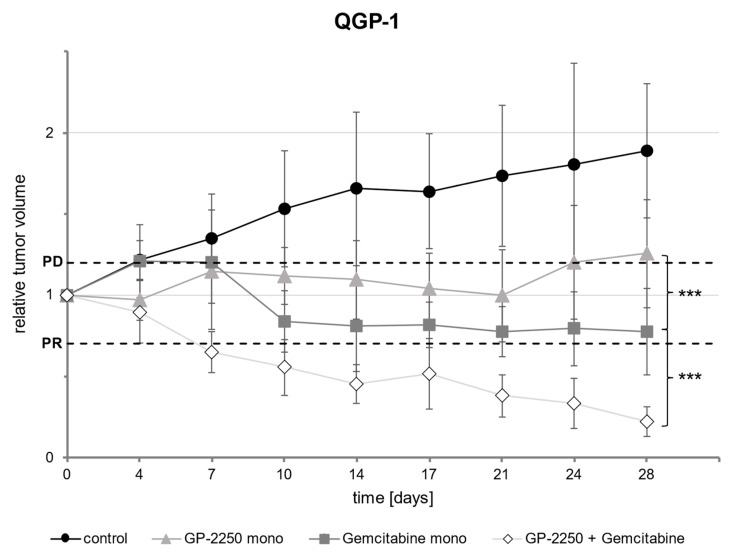
Development of the relative tumor volume of QGP-1 during the observation period. Figure 6 shows the development of the relative tumor volume during the observation period of Bo99 tumors treated with Gemcitabine mono and a combination of Gemcitabine and GP-2250 in comparison to the untreated controls. Error bars show the standard deviation. *p*-values are indicated as follows: *** *p* ≤ 0.001.

## Data Availability

The datasets used and analyzed during the current study are available from the corresponding author on reasonable request.

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
