# Peer review of "New Therapy Options for Neuroendocrine Carcinoma of the Pancreas—The Emergent Substance GP-2250 and Gemcitabine Prove to Be Highly Effective without the Development of Secondary Resistances In Vitro and In Vivo"

_cancers, 2022, doi:10.3390/cancers14112685_

Round 1
Reviewer 1 Report
I commend the authors: Marie Buchholz, Johanna Strotmann, Britta Majchrzak-Stiller , Stephan Hahn, Ilka Peters , Julian Horn , Thomas Müller, Philipp Höhn, Waldemar Uhl and Chris Braumann of the manuscript titled “New therapy options for neuroendocrine carcinoma of the pancreas - emergent substance GP-2250 and Gemcitabine proves to be highly effective without development of secondary resistances in vitro and in vivo” for their work on the effects of the emerging substance GP-2250 on pNEC.
Before this manuscript is published, there are several things need to be addressed or corrected:
- The English in general need to be improved.
- In the introduction:
- The introduction is very short and additional review should be added showing current status, importance of the work, the oxathiazinane derivate GP-2250..etc.
- Show what other materials used to control these cancers.
- You need to add the novelty of the work here especially in the last paragraph and not single sentence as you do.
- In the material and methods:
- section 2.3. : Citation and references for all the methods used in the section should be added. You are not the first one to do that, you followed other researcher whom should be cited.
- The same comment applies to section 2.4.
- In the results and discussion
- Figures 2-5 should be enlarged, sharpened and enhanced.
- p ≤ 0.001, ** p ≤ 0.01, * p ≤ 0.05 should be italic
- In the discussion
- additional references and talk should be added about comparing your results with other materials used to control the studied tumor showing the importance of data and if its higher or lower and explanations.
- The conclusion, you need to add the prospect of this work. Add the limitation of the study as well.
Author Response
Manuscript ID cancers- 1722022
Response to Reviewer 1
Dear Reviewer 1,
Thank you for giving us the opportunity to submit a revised draft of the manuscript “ New therapy options for neuroendocrine carcinoma of the pancreas – emergent substance GP-2250 and Gemcitabine proves to be highly effective without development of secondary resistances in vitro and in vivo ” for publication in the Journal Cancers.
We appreciate the time and effort that you dedicated providing feedback on our manuscript and are grateful for the insightful comments on and valuable improvements to our paper.
We have incorporated the suggestions made. Those changes are highlighted within the manuscript. Please see below, in blue, for a point-by-point response to your comments and concerns. All page numbers refer to the revised manuscript file with tracked changes.
- The English in general need to be improved.
Authors response: Thank you for your remarks concerning the English language. We revised the English language carefully and altered it to the more commonly used american spelling
- In the introduction: The introduction is very short and additional review should be added showing current status, importance of the work, the oxathiazinane derivate GP-2250.etc.
Show what other materials used to control these cancers.
You need to add the novelty of the work here especially in the last paragraph and not single sentence as you do.
Authors response: Thank you for pointing this out. We have reviewed the introduction section completely. We added more detailed information about GP-2250 and other agents used in in the therapy of the neuroendocrine carcinoma. We pointed out the novelty of the work more comprehensively. Find the revised section from line 58 to line 92.
- In the material and methods: section 2.3. : Citation and references for all the methods used in the section should be added. You are not the first one to do that, you followed other researcher whom should be cited. The same comment applies to section 2.4.
Authors response: Thank you for your comment. We added references for all the methods used in this study.
- In the results and discussion: Figures 2-5 should be enlarged, sharpened and enhanced.
p ≤ 0.001, ** p ≤ 0.01, * p ≤ 0.05 should be italic
Authors response: Thank you for your suggestion. We enlarged, sharpened and enhanced the figures. Please find the revised figures in line 207, line 235, line 354, 283, line 299 and line 319. The p-values were set to italic.
- In the discussion: additional references and talk should be added about comparing your results with other materials used to control the studied tumor showing the importance of data and if its higher or lower and explanations.
Authors response: Thank you for pointing this out. We revised the discussion section, included additional references and talk and compared our results with the current state of research in pancreatic neuroendocrine cancer. Find the revised section from line 329 to line 351, line 368 to line 382 and line 392 to line 397.
- The conclusion, you need to add the prospect of this work. Add the limitation of the study as well.
Authors response: Thank you for your comment. We added the limitations of our study and the prospects of this work. Find the revised section from line 401 to line 409.

Reviewer 2 Report
The manuscript titled : New therapy options for neuroendocrine carcinoma of the pancreas - emergent substance GP-2250 and Gemcitabine proves to be highly effective without development of secondary resistances in vitro and in vivo” is interesting work and should be published after minor revision.
Before this manuscript is published, there are few things need to be addressed or corrected:
- The English in general need to be little bit improved.
- Additional literatue should be added to the introduction section to improve the quality of the presentation of your interesting and novel work.
- Please add citation to the methods your group used.
- In the results and discussion, the figures in general need to enlarged and enhanced.
- Please add the limitations of the study
Author Response
Manuscript ID cancers- 1722022
Response to Reviewer 2
Dear Reviewer 2,
Thank you for giving us the opportunity to submit a revised draft of the manuscript “ New therapy options for neuroendocrine carcinoma of the pancreas – emergent substance GP-2250 and Gemcitabine proves to be highly effective without development of secondary resistances in vitro and in vivo ” for publication in the Journal Cancers.
We appreciate the time and effort that you dedicated providing feedback on our manuscript and are grateful for the insightful comments on and valuable improvements to our paper.
We have incorporated the suggestions made. Those changes are highlighted within the manuscript. Please see below, in blue, for a point-by-point response to your comments and concerns. All page numbers refer to the revised manuscript file with tracked changes.
- The English in general need to be a little bit improved.
Authors response: Thank you for your remarks concerning the English language. We revised the English language and altered it to the more commonly used American spelling.
- Additional literature should be added to the introduction section to improve the quality of the presentation of your interesting and novel work.
Authors response: Thank you for pointing this out. We have reviewed the introduction section completely. We added literature and more detailed information about GP-2250 and the current therapy of the neuroendocrine carcinoma. We pointed out the novelty of the work more comprehensively. Find the revised section from line 58 to line 92.
- Please add citation to the methods your group used.
Authors response: Thank you for your comment. We added references for the methods used in this study.
- In the results and discussion, the figures in general need to enlarged and enhanced.
Authors response: Thank you for your suggestion. We enlarged and enhanced the figures. Please find the revised figures in line 207, line 235, line 354, 283, line 299 and line 319.
- Please add the limitations of the study
Authors response: Thank you for your comment. We added the limitations of our study. Find the revised section from line 388 to line 393 and from line 407 to line 408.

Round 2
Reviewer 1 Report
Accepted for me
This manuscript is a resubmission of an earlier submission. The following is a list of the peer review reports and author responses from that submission.